

# Unexpected long-range transport of glyoxal and formaldehyde observed from the Copernicus Sentinel-5 Precursor satellite during the 2018 Canadian wildfires

Leonardo M. A. Alvarado[1], Andreas Richter[1], Mihalis Vrekoussis[1,2,4], Andreas Hilboll[1], Anna B. Kalisz Hedegaard[3,1], Oliver Schneising[1], and John P. Burrows[1]

[1]Institute of Environmental Physics (IUP), University of Bremen, Bremen, Germany
[2]Energy, Environment, and Water Research Center, The Cyprus Institute, Nicosia, Cyprus
[3]Institute of Atmospheric Physics, German Aerospace Center (DLR), Oberpfaffenhofen-Wessling, Germany
[4]Center of Marine Environmental Sciences (MARUM), University of Bremen, Bremen, Germany

**Correspondence:** Leonardo M. A. Alvarado (lalvarado@iup.physik.uni-bremen.de)

**Abstract.** Glyoxal (CHO.CHO) and formaldehyde (HCHO) are intermediate products in the oxidation of the majority of volatile organic compounds (VOC). CHO.CHO is also a precursor of secondary organic aerosol (SOA) formation in the atmosphere. These VOCs are released from biogenic, anthropogenic, and pyrogenic sources. CHO.CHO and HCHO tropospheric lifetimes are short during the daytime and at mid-latitudes (few hours), as they are rapidly removed from the atmosphere by

their photolysis, oxidation by OH, and uptake on particles/deposition. During nighttime or at high latitudes, lifetime can be prolonged to many hours or even days. Previous studies demonstrated that CHO.CHO and HCHO can be retrieved from space-borne observations using the DOAS method. In this study, we present CHO.CHO and HCHO columns retrieved from measurements of the TROPOMI instrument, launched recently on the Sentinel-5 Precursor (S5P) platform in October 2017. Strongly elevated amounts of CHO.CHO and HCHO are observed during the fire season in British Columbia, Canada, where

a large number of fires occurred in August 2018. CHO.CHO and HCHO plumes from individual fire hot-spots are observed in air masses travelling over distances of up to 1500 km, i.e. much longer than expected for the short atmospheric lifetime of CHO.CHO and HCHO. However, the temporal evolution of the plume differs for both species. Comparison with Lagrangian-based FLEXPART simulations for particles with different lifetimes shows that effective lifetimes of 20 hours and more are needed to explain the observations, indicating that CHO.CHO and HCHO are either efficiently recycled during transport or,

continuously formed from the oxidation of longer-lived precursors present in the plume.

## 1 Introduction

Formaldehyde (HCHO) is produced in the oxidation of both methane ($CH_4$) and other Volatile Organic Compounds (VOC). Glyoxal (CHO.CHO) is the smallest alpha-dicarbonyl formed in the oxidation of most VOC containing two or more carbon atoms. Although both HCHO and CHO.CHO, which are known as OVOC (Oxygenated Volatile Organic Compounds) have

similar rates of reaction with the hydroxyl radical (OH) in the troposphere, the photolysis frequency of HCHO, which absorbs and is photolysed in the ultraviolet-A (UV-A), is significantly smaller than that of CHO.CHO, which absorbs in the blue. As





a result, the atmospheric lifetime of HCHO is longer than that of CHO.CHO (Atkinson, 2000). Both species are short-lived during daytime due to their rapid removal by photolysis and reaction with OH radicals (Atkinson, 2000; Volkamer et al., 2007). These processes are the major sinks of CHO.CHO and HCHO contributing about 69% and 96%, respectively, the

remaining part of HCHO being removed by deposition (4%), while for CHO.CHO 22% is removed by SOA formation and 8% by deposition (Stavrakou et al., 2009a, c). Additionally, HCHO during nighttime can also be removed by reaction with nitrate ($NO_3$) radicals (Atkinson, 2000). HCHO and CHO.CHO play a key role in tropospheric chemistry because they act as temporary reservoirs of VOC; additionally, they produce carbon monoxide (CO) and OHx (OH and Hydroperoxyl, $OH_2$) free radicals, which participate in catalytic cycles creating and destroying tropospheric ozone ($O_3$). The column amounts of HCHO

were first observed from space using measurements from the GOME instrument (e.g. Burrows et al., 1999, and references therein). These columns were later used to estimate the emission strength of precursor VOC (Palmer et al., 2003; Abbot et al., 2003). The simultaneous observation of CHO.CHO and HCHO (Wittrock, 2006) enabled an improved assessment of atmospheric VOC levels and the ratio of CHO.CHO-to-HCHO (RGF), (Vrekoussis et al., 2010), provided some source differentiation. Studies have used HCHO, partly in combination with CHO.CHO to estimate the biogenic isoprene emissions

(Fu et al., 2007; Stavrakou et al., 2009a, b, c; Liu et al., 2012; Marais et al., 2012). This is the largest natural source of CHO.CHO (Guenther et al., 2006; Fu et al., 2007). The amount of biogenically emitted VOC depends on several factors including, among others, the plant species and weather conditions (e.g. temperature and humidity) (Guenther et al., 2000). In industrialized areas, there are also contributions to the amounts of CHO.CHO from human activities, such as from fossil fuel production, distribution and combustion: the largest source of VOC precursors of CHO.CHO being motor vehicle emissions

due to either evaporation or incomplete combustion of fuel (Kansal, 2009). Globally, 55% of CHO.CHO is produced by biogenic precursors, while 27% are from anthropogenic and the remaining 18% from pyrogenic emissions, which include wildfires and biomass burning (Stavrakou et al., 2009a). Fires and vehicle exhausts are thought to be the only two sources, which directly emit CHO.CHO (Stavrakou et al., 2009a; Zhang et al., 2016). In August 2018, unusualy high temperatures caused severe drought in some areas of North America and resulted in the outbreak of many wildfires: the province of British

Columbia (BC) in Canada was one of the most affected areas. The 2018 season has been the worst on record, with 6826 fires being detected and an area of approximately $22500\,km^2$ of land burned (Canada, 2018). These fires emitted many different species into the atmosphere, e.g. CO, $NO_x$, VOC, OVOC, $O_3$, $SO_2$, $CO_2$, HCHO, HONO, $CH_3CO.O_2.NO_2$ (PAN) and other toxic species and aerosols (Urbanski et al., 2018). During the transport of plumes from fires, photochemical transformation of emitted species occurs. Overall pollution resulting in low air quality is transported to those regions where the plumes descend

to the surface. HCHO and CHO.CHO column amounts are observed by remote sensing using Differential Optical Absorption Spectroscopy, (DOAS), using measurements of the radiances backscattered from the Earth's surface and atmosphere. The global maps of the HCHO and CHO.CHO, retrieved from SCIAMACHY, GOME-2, and OMI show enhanced HCHO and CHO.CHO over tropical rain forests but also over other regions with high isoprene emissions. In addition, hot-spots of HCHO and CHO.CHO from fire emissions can be detected over large wildfires (Wittrock et al., 2006; Vrekoussis et al., 2009, 2010;

Lerot et al., 2010; Chan Miller et al., 2014; Alvarado et al., 2014; De Smedt et al., 2008, 2012, 2015; Smedt et al., 2018). In this study, we present novel observations of CHO.CHO retrieved from the high spatial resolution observations of the instrument





TROPOMI on board the S5P platform. On August 07, 2018, strongly elevated amounts of CHO.CHO and HCHO were observed over British Columbia and attributed to being predominantly from the fires. Surprisingly, these elevated levels of CHO.CHO and HCHO were not limited to the vicinity of the fires. The fire plumes, which contain both CHO.CHO and HCHO remain visible for several days and appear to travel long distances from the sources. The observations, in combination with forward simulations of atmospheric transport calculated using the FLEXPART model (Pisso et al., 2019a), enable the investigation of long-range transport of CHO.CHO and HCHO during the studied period. FLEXPART simulations of a tracer emitted over the fire hot-spots and having a lifetime of $\approx$29 hours can reproduce the evolution of the plumes of CHO.CHO and HCHO for most of the fire events, and thus provide estimates of the effective lifetimes of CHO.CHO and HCHO in the plumes as is described in the sections below. In addition, carbon monoxide (CO) retrievals also from the TROPOMI instrument and true color images from VIIRS satellite are used as complementary information in order to interpret the enhanced lifetime of CHO.CHO and HCHO in the plume. Additionally, interpretation of the main source of these species is performed by computing the ratio of CHOCHO to HCHO for the specific events.

## 2 Methods

### 2.1 CHO.CHO and HCHO observations

The Differential Optical Absorption Spectroscopy (DOAS) method has been successfully applied to retrieve atmospheric columns of trace gases having fingerprint narrow absorption bands in the solar spectral range from space-borne instruments (e.g. Burrows et al., 1999). As noted above, there are several studies describing retrievals of OVOC and or their use for VOC sources estimation (Burrows et al., 1999; Palmer et al., 2001; Wittrock et al., 2006; Kurosu et al., 2007; Vrekoussis et al., 2009, 2010; Lerot et al., 2010; De Smedt et al., 2008, 2012, 2015; Smedt et al., 2018; Hewson et al., 2013; González Abad et al., 2015; Chan Miller et al., 2014; Alvarado et al., 2014, 2015). Algorithms for the retrieval of HCHO and CHO.CHO have been developed for measurements from the SCanning Imaging Absorption spectroMeter for Atmospheric CHartographY (SCIAMACHY) (Burrows et al., 1995; Bovensmann et al., 1999), the Ozone Monitoring Instrument (OMI) (Levelt et al., 2006), and the second Global Ozone Monitoring Experiment on MetOp–A and –B (GOME2–A and–B) (Munro et al., 2016), which in combination provide a continuous dataset covering a period of more than 20 years. In this study, measurements from the TROPOMI instrument on board of the Sentinel 5 precursor (Veefkind et al., 2012) are used to retrieve atmospheric column amounts of CHO.CHO and HCHO. A brief instrument description and relevant details of the retrieval of CHO.CHO and HCHO are given below.

### 2.2 The TROPOMI instrument

The TROPOspheric Monitoring Instrument (TROPOMI) onboard of the Copernicus Sentinal-5 Precursor satellite was launched on 13 October 2017. It has a spectral range in the UV-VIS-NIR-SWIR covering wavelengths from 270 to 500 nm in the UV-VIS, from 675 to 775 nm in the NIR and in a SWIR band from 2305 to 2385 nm. These bands allow the observation of





several relevant atmospheric species, including CHO.CHO, HCHO, NO$_2$ and CO. TROPOMI provides nearly global coverage each day at a spatial resolution of currently 3.5 km×7 km (7 km×7 km in the SWIR). The equator crossing time is 13:00 LT

(ascending node). Similar to OMI, TROPOMI is a nadir-viewing imaging spectrograph, which consists of a two-dimensional CCD, one dimension collecting the spectral information, the other being used for the spatial information. The TROPOMI instrument onboard of the S5P satellite provides data since November 2017 (Veefkind et al., 2012).

### 2.3 CHO.CHO retrieval from TROPOMI measurements

In recent years, several improvements on the retrieval of CHO.CHO have been reported. In 2014, Chan Miller et al. (2014) and

(Alvarado et al., 2014) presented new CHO.CHO retrieval algorithms applied to OMI measurements. These studies, similar to previous studies on GOME–2A data, introduced approaches to reduce interference by absorbers, such as liquid water and nitrogen dioxide (NO$_2$). In this study, an optimized retrieval algorithm for CHO.CHO was developed, building on the heritage from the OMI CHO.CHO retrieval presented by (Alvarado et al., 2014), extended and applied to S5P measurements. Previous studies have shown that cross-correlations between references cross-sections, as well as instrumental structures or features and

shifts in the wavelength calibration can introduce systematic errors in the retrieval. As a result, a strong dependence on the fitting window was identified in the retrieved CHO.CHO slant column densities, SCDs (Chan Miller et al., 2014; Alvarado et al., 2014). In this study, a fitting window from 433 to 465 nm is chosen, which is slightly larger than windows used in previous investigations (Vrekoussis et al., 2010; Alvarado et al., 2014). This fitting window, which enables the liquid water absorption to be retrieved, leads to a reduction in the number of negative CHO.CHO SCs over oceanic regions in comparison

to a shorter fitting window (e.g. 434–458 nm), as well as a reduction in the residuals. The wavelength range selected covers the strong absorption bands of CHO.CHO (452-457 nm), which have already been used in the past to retrieve CHO.CHO from ground and ship-based DOAS configurations as well as from satellites (Sinreich et al., 2007, 2010; Wittrock et al., 2006; Vrekoussis et al., 2009, 2010; Lerot et al., 2010; Chan Miller et al., 2014; Alvarado et al., 2014). In order to optimize the quality of the retrievals, a row-dependent daily mean Pacific spectrum from the region 50° S, 160° E - 50° N, 135° W is used

as background spectrum (Alvarado, 2016). In addition, the mean CHO.CHO SC over the region 30° S, 150° W – 30° N, 150° E is computed each day and subtracted from all SC to correct for possible offsets. A summary of the selected absorption cross-sections, and other parameters used in the retrieval, as well as a list of the species included in the retrieval, is shown in Table 1. SCDs depend on observation geometry. Vertical column densities (VCD) are derived from the SCDs by use of so-called air mass factors (AMF), which depend on the trace gas profile, air pressure, surface albedo, temperature, aerosols,

clouds and on solar zenith angle and measurement geometry. As the focus of this study is the observation of CHO.CHO in biomass burning emissions, a simple CHO.CHO profile with a Gaussian distribution having its maximum peak at the altitude of the aerosol layer is used (see Figure 1-A). This is based on the assumption that CHO.CHO is found at the same location as the main plume of aerosol and other trace gases. The altitude of the aerosol layer was estimated from profiles retrieved by the Cloud-Aerosol Lidar and Infrared Pathfinder Satellite Observation (CALIPSO) (Vaughan et al., 2004) (Figure 1-B). These

aerosol extinction coefficients (k$_{ext}$) profiles retrieved at 532 nm are also used in the calculation of the AMFs by the radiative transfer model SCIATRAN (Rozanov et al., 2013). The computations have been performed on a daily basis, assuming a single





**Table 1.** Summary of retrieval parameters of CHO.CHO and HCHO de from S5P with the respective absorption cross-sections used.

| Parameters | Formaldehyde (HCHO) | Glyoxal (CHO.CHO) |
|---|---|---|
| Fitting window | 323.5-361 nm | 433-465 nm |
| Polynomial | 5 coefficients | 5 coefficients |
| Cross-sections used: | | |
| HCHO (Meller and Moortgat, 2000) | Yes (298 K) | No |
| CHO.CHO (Volkamer et al., 2005b) | No | Yes (296 K) |
| $NO_2$ (Vandaele et al., 1998) | Yes (220 K) | Yes (220 K, 294 K) |
| O4 (Thalman and Volkamer, 2013) | Yes (293 K) | Yes (293 K) |
| $O_3$ (Serdyuchenko et al., 2014) | Yes (223 K, 243 K) | Yes (223 K) |
| BrO (Fleischmann et al., 2004) | Yes (223 K) | No |
| $H_2O$ (Rothman et al., 2013) | No | Yes (296 K) |
| Liquid water (Mason et al., 2016) | No | Yes (280 K) |
| Ring effect | Ring cross section calculated by SCIATRAN model Vountas et al. (1998) | |
| Non-linear ozone absorption effects, 2 pseudo absorption cross-sections ($O_3 * \lambda + (O_3)^2$) from Taylor expansion (Puķīte et al., 2010) | Yes | No |
| Iterative spike removal (Richter et al., 2011) | Applied | |
| Intensity offset correction | Linear offset ($I/I_0$) | |
| Background spectrum | Pacific region ($50° N, 135° W – 50° S, 160° W$) | |

scattering albedo of 0.92 and a homogenous distribution of aerosols characterised by the mean profile in the whole region of study. The latter is computed from the average of all aerosol profiles, for every single latitude and longitude, and by removing cloud-contaminated pixels (see Figure 1-C). Clouds are not explicitly accounted for in the CHO.CHO and HCHO retrievals

but data are filtered for the presence of clouds using an intensity criterion corresponding to a cloud radiance fraction of about 50%.

## 2.4 HCHO retrieval from TROPOMI

The accuracy of DOAS retrievals of HCHO is limited by cross-correlations with strong absorbers in the UV (e.g. $O_3$) and the signal to noise ratio of the radiance spectra measured by the instrument. Here, an updated version of the formaldehyde

retrieval developed by Wittrock et al. (2006) and Vrekoussis et al. (2010) is used, which applies a slightly larger fitting window extending from 323.5 nm to 361 nm, resulting in a reduction in the noise of the retrieved slant columns. At wavelengths shorter than 336 nm, interference with $O_3$ is observed due to the strong absorption of the latter. This effect is compensated by applying the method described by Puķīte et al. (2010), which consist of adding two additional pseudo-cross-sections to the fit ($\lambda\sigma_{O_3}$ and $\sigma_{O_3}^2$) (Puķīte et al., 2010; De Smedt et al., 2008, 2015; Smedt et al., 2018). The cross-sections of interfering species are





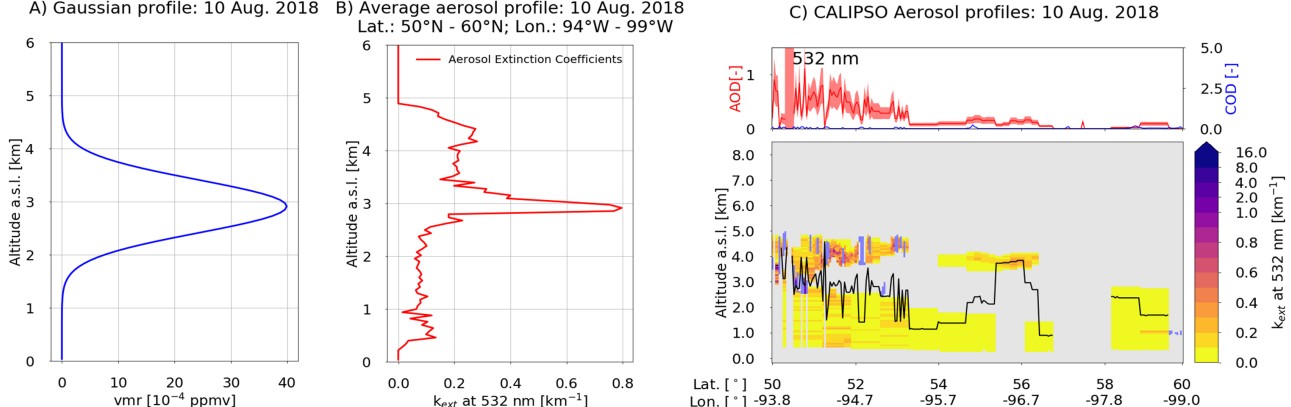

**Figure 1.** A) CHO.CHO and HCHO profiles assumed in the computation of AMFs. B) CALIPSO average profile of aerosol extinction coefficients (kext) for all latitudes and longitudes of Figure 1-C, excluding cloudy scenes. C) Top panel: Example of CALIPSO Aerosol profile extinction coefficients retrieved at a wavelength of 532 nm. Aerosol and cloud optical depth are shown as a function of latitude and longitude for every single profile. Bottom panel: Colour coded kext for every latitude and longitude in the selected region. Purple spots represent cloudy scenes. The black line depicts the aerosol layer height.

included in the fit as listed in Table 1. Similar to glyoxal, a synthetic ring spectrum (Vountas et al., 1998) is used to account for the Ring effect and a row-dependent daily mean Pacific spectrum from the region $50°$ S, $135°$ W - $50°$ N, $160°$ E is used as background spectrum. Also, a latitude dependent offset correction based on SCDs from longitudes between $180°$ E and $160°$ E is applied to the data. As for CHO.CHO, vertical columns are computed using AMFs, assuming a Gaussian shape for the distribution of HCHO at the layer where the aerosols are located. Figures 2-A and 2-B show examples of CHO.CHO and

HCHO fit results for the 10th of August 2018, compared to the differential reference cross-section for a single measurement. For an individual CHO.CHO measurement, the detection limit is of the order of $5 \times 10^{14}$ molec.cm$^{-2}$, which is about 10 times smaller than the columns detected from emissions of the wildfires over the British Columbia region of Canada. For HCHO, the detection limit is an order of magnitude higher ($5.5 \times 10^{15}$ molec.cm$^{-2}$). The detection limit of a single S5P measurement in this study has been estimated in a manner similar to that explained in Alvarado et al. (2014).

**2.5   Simulation of tracer transport with FLEXPART**

In order to simulate the transport of emissions from the Canadian wildfires, forward simulations with version 10.3 of the FLEXible PARTicle dispersion model FLEXPART (Stohl et al., 2005; Pisso et al., 2019b) have been performed. The model was driven by using hourly wind fields from the ECMWF ERA5 reanalysis (C3S) at $0.250°$ horizontal resolution. As a transport model, FLEXPART cannot adequately simulate the chemical transformations leading to the observed lifetimes of trace gases in

the biomass burning plumes. Nevertheless, performing simulations for tracers having different mean lifetimes and comparing these to the observed columns yields valuable insight into the effective lifetime, which the emitted substances have to have





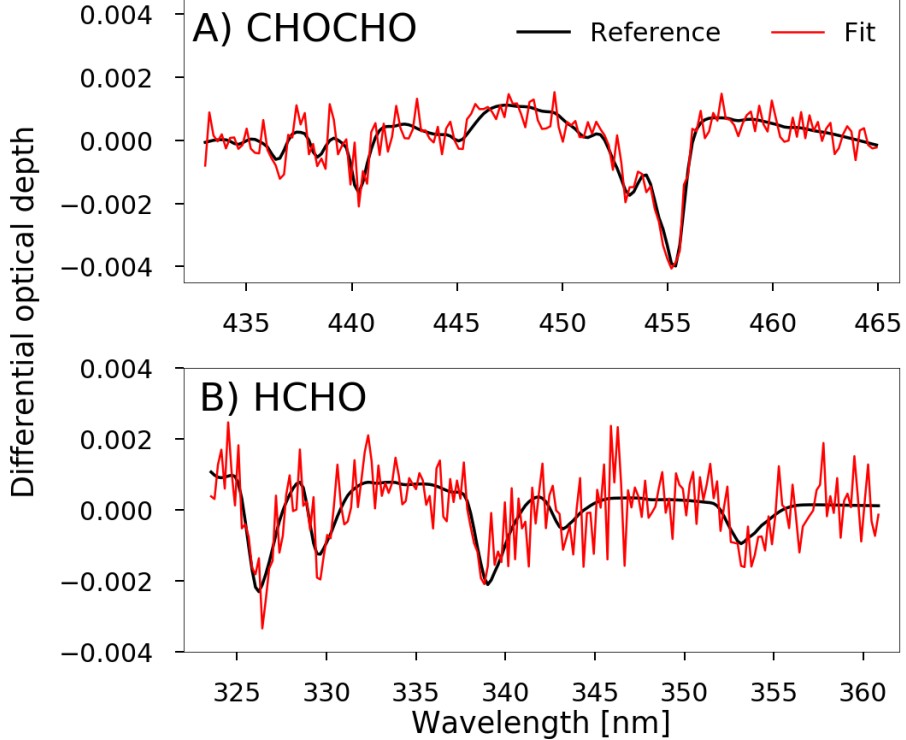

**Figure 2.** A) Example fit for CHO.CHO from a single measurement of S5P taken at latitude 53.0° and longitude 125.6° W, on the 10th of August 2018 and for a solar zenith angle of 39.3°. B) Example fit for HCHO from a single measurement of S5P taken at latitude 59.1° and longitude 109.0° W, on the 10th of August 2018 and for a solar zenith angle of 44.6°. The black line depicts the scaled differential cross-section and the red line the fit. The SCD values for this example are $9.3 \times 10^{15}$ molec.cm$^{-2}$ for CHO.CHO and $4.6 \times 10^{16}$ molec.cm$^{-2}$ for HCHO, respectively. The detection limit for a single measurement from S5P is estimated to be $5.0 \times 10^{14}$ molec.cm$^{-2}$ and $5.5 \times 10^{15}$ molec.cm$^{-2}$ for CHO.CHO and HCHO, respectively.

in the plume, in order to explain the observed plume evolution. In FLEXPART, the mean effective lifetime $\tau$ of an emitted tracer is treated as exponential decay with a given half-life ($t_{0.5}$); $\tau$ can then be calculated according to $\tau = t_{0.5}/ln(2)$. As part of this study, FLEXPART simulations were carried out with half-life times of 2, 4, 6, 8, 10, 12, 14, 16, 18, and 20 hours, corresponding to effective mean lifetimes of $\sim$2.9, 5.8, 8.7, 11.5, 14.4, 17.3, 20.2, 23.1, 26.0, and 28.9 hours, respectively. As the exact emissions from the wildfires are highly uncertain, emission fluxes from the wildfires are assumed to be proportional to fire radiative power (FRP, see below for more details). This means that effectively, emissions are prescribed in the model in arbitrary mass units; scaling the individual emission sources with FRP ensures that when aggregating the simulation results from all fires, each fire's relative contribution to the simulated columns is retained, and the results can be compared to the observed columns. The emissions, prescribed in the model, are taken from the Global Fire Assimilation System (GFAS) daily FRP and plume height data Rémy et al. (2017). Simulations were performed on a daily basis for the period 06 Aug to 23





Aug 2018. For each day, all fires from the GFAS data, which had an FRP of more than $3\,\mathrm{W/cm^{-2}}$ were gridded to a $0.350°$ horizontal pattern. The model was then run forward in time for 120 hours, releasing the tracer for the first 24 hours (the full UTC day) from each of the $0.350°$ grid cells, assuming no temporal variation throughout the day. Vertically, the emissions within the grid cells were evenly distributed over the range of mean altitude of maximum injection heights given by the GFAS data for the respective grid cell. The output of the simulation contains gridded mass concentrations for each time step. Here, a grid with a horizontal resolution of $0.031250°$ was chosen, to match the resolution of the gridded satellite observations. Hourly output from the simulation was recorded and then vertically integrated to yield simulated tracer columns. In a post-processing step, for one specific mean lifetime, all simulation results (i.e., simulations for all fires on all days) were aggregated into one dataset. While the absolute tracer column density from the model output cannot be simply compared to the measurements, a comparison of the plume patterns and relative distribution between satellite observation and model output gives an indication about the meaningfulness of the prescribed mean lifetime. At this point, the aggregated model output for one effective mean lifetime consists of hourly latitude-longitude grids of vertical tracer columns throughout the whole study period. For comparison to the satellite observations, the hourly time slice closest to the time of overpass at $53°$ N was chosen.

## 3   Results and discussion

During August 2018, a high-temperature anomaly led to the outbreak of many fires in the Canadian Western province of British Columbia, resulting in the emission of large quantities of particles and traces gases that in turn affected air quality in the region. As shown in Figure 3-A and -B, the monthly average of CHO.CHO and HCHO vertical columns from S5P show strongly enhanced values over the fire region, suggesting that these fires were a large direct and/or indirect source of CHO.CHO and HCHO. Surprisingly, the CHO.CHO and HCHO enhancements are not limited to the main fire region but extend over large parts of Canada, where only a few fires were observed. In order to investigate the sources of CHO.CHO and HCHO and their distributions, 24-hour assimilation data of fire radiative power from the Global Fire Assimilation System (Kaiser et al., 2012) are analysed. Briefly, FRP is a measure of outgoing radiant heat from fires, measured in units of $\mathrm{W.cm^{-2}}$ and retrieved from space by the MODerate resolution Imaging Spectroradiometers (MODIS) on board of Terra and Aqua satellites (Justice et al., 2002). The assimilated FRP spatially aggregates all valid fire and non-fire observations from both MODIS instruments onto a horizontal resolution of $0.1° \times 0.1°$ and computes the total FRP sums for each grid bin (Justice et al., 2002). The FRP is also used as input in the FLEXPART simulation as described in section 2.5 as a proxy for emission strength. Figure 3-C shows a monthly average FRP map over North America for August 2018.

The highest CHO.CHO values are found over the locations of the most intense fires, as intuitively expected. The HCHO distribution over the fire regions is similar to that of CHO.CHO, but with some differences in the relative distribution. In addition, enhanced HCHO columns are also apparent over the south-eastern US, where large isoprene emissions occur. CHO.CHO and HCHO are also detected in plumes crossing central and eastern Canada, where no fires are identified in the FRP map. This pattern points to transport of CHO.CHO and HCHO emanating from the wildfires. However, it is well-known that CHO.CHO and HCHO have short atmospheric lifetimes of about $\sim2.2$ and $\sim4.0$ hours during daytime, respectively (Atkinson, 2000;



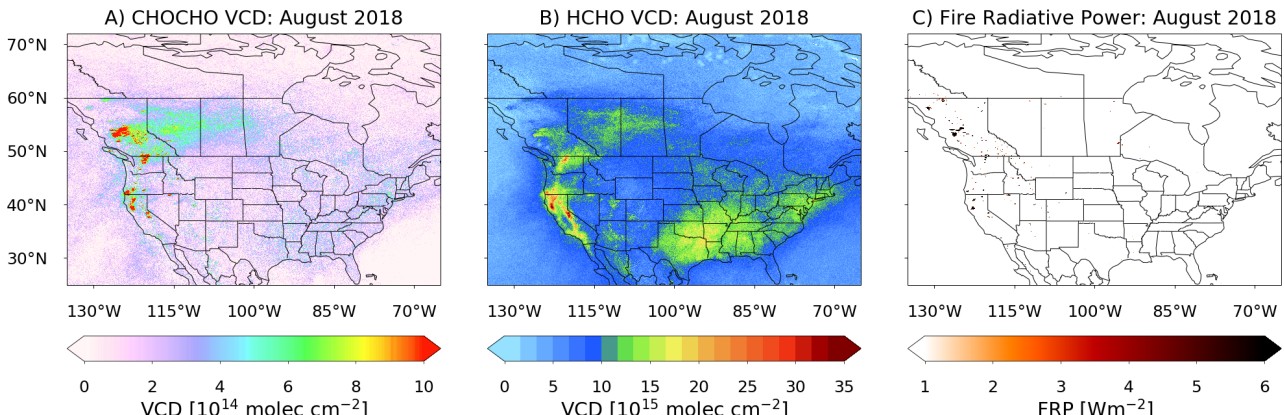

**Figure 3.** Monthly average of CHO.CHO (panel A) and HCHO (panel B) VCDs retrieved from the TROPOMI instrument for August 2018, and over North America (A and B). Panel C shows the integrated FRP from MODIS for the same period.

Volkamer et al., 2005a), which is not sufficient to explain the observed pattern. Earlier studies by Wittrock et al. (2006) and Vrekoussis et al. (2009, 2010) showed that CHO.CHO is also observed over oceanic regions, where no CHO.CHO source is expected. The potential of a) a long-range transport of CHO.CHO and/or of CHO.CHO precursors from continental areas, and b) having an unknown oceanic CHO.CHO source were discussed as a possible explanation of these observations, but no firm conclusions could be drawn so far. In the present study, with the support of the FLEXPART model, evidence of long-range

transport of CHO.CHO and HCHO or its precursors from biomass burning emissions is shown. In order to further investigate the transport of CHO.CHO and HCHO, two different periods of satellite measurements are selected (07-10 and 20-21 of August 2018) and discussed below.

### 3.1 CHO.CHO and HCHO emissions from the British Columbia wildfires: 07-10 and 20-21 August 2018

Figure 4-A shows daily maps of CHO.CHO and HCHO VCDs over Canada for the period 7th to 10th of August 2018. The

most intense wildfires are found on the 7th of August 2018 and remain detectable until the 10th of August. The location of those fires corresponds well to the location, where both trace species are detected on the first day. The CHO.CHO and HCHO distributions are changing from day to day. However a large plume is formed on the 10th of August 2018. Enhanced CHO.CHO and HCHO columns are found at a distance of up to ∼1500 km from the fires, indicating transport over long distances. To investigate possible transport pathways, forward simulations of the atmospheric transport with FLEXPART were

calculated for the same period as HCHO and CHO.CHO observations (see Figure 4-B), assuming a random effective lifetime of 14.4 hours. The latter is in contrast to the lifetimes of CHO.CHO and HO expected, which are significantly shorter. On the other hand, the pattern of air masses follows the distribution of CHO.CHO and HCHO columns well, showing good spatial agreement between simulations and satellite observations. The tracer simulated with FLEXPART spreads over the same area as CHO.CHO, providing evidence for the transport of CHO.CHO and HCHO and their precursors over continental Canada. This





is even more evident for the second period of interest in this study, which extends from the 20th to 21st of August 2018 (see
Figure 7). While the spatial match of plume and model is good in this example, it is clear from the figure that even an effective
lifetime of 14.4 hours does not describe fully the extent of the trace gas transport, and using shorter lifetimes for CHO.CHO
and HCHO would not reproduce the observations, however both lifetimes depend on time of day (daytime and nighttime) and
also on photon flux and OH diurnal pattern. Consequently, comparisons of FLEXPART simulations with different effective
lifetimes were performed for two selected days, which are detail discussed in Section 3.2.

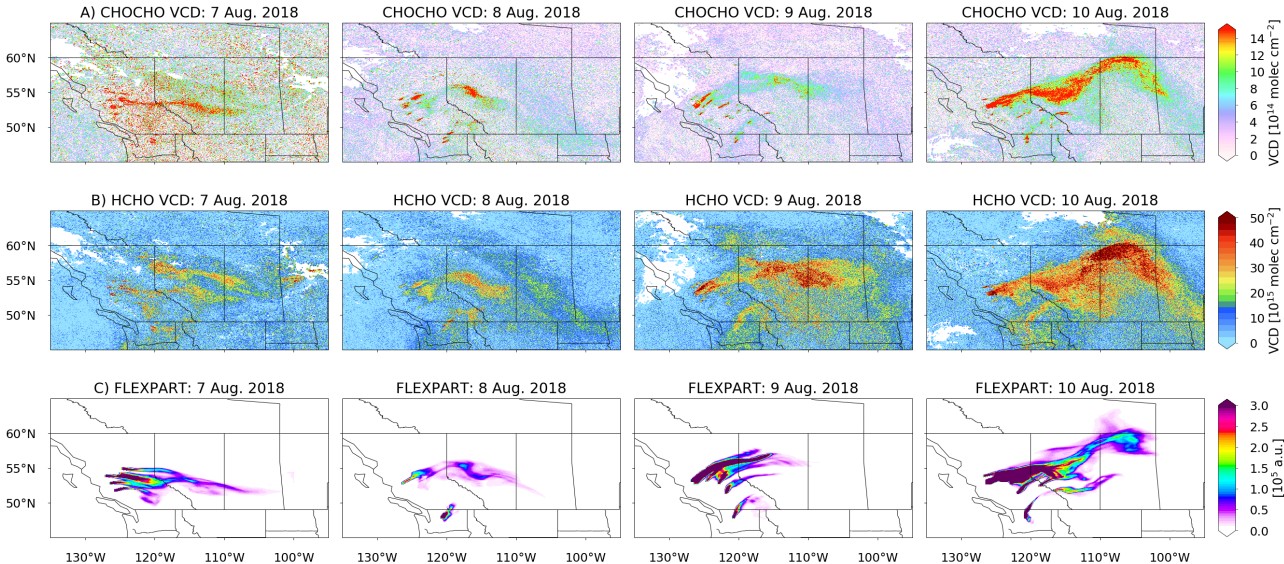

**Figure 4.** A) and B) Daily CHO.CHO and HCHO VCDs retrieved from S5P measurements for the period 7 to 10 of August. C) Distribution
of a tracer with a lifetime of 14.4 hours simulated with FLEXPART for the same period. The CHO.CHO in the plume decreases on average
from $3 \times 10^{15}$ molec.cm$^{-2}$ to $3 \times 10^{14}$ molec.cm$^{-2}$, while the HCHO has a different variation into the plume but at the end of the plume,
it decreases from $4 \times 10^{16}$ molec.cm$^{-2}$ to $2 \times 10^{16}$ molec.cm$^{-2}$. The FLEXPART tracer column decreases from $3 \times 10^{6}$ to $0.3 \times 10^{6}$ for
this specific effective lifetime of 14.4 hours.

## 3.2  Lifetimes of CHO.CHO and HCHO in the plume

Figure 5 shows the results of FLEXPART simulations assuming effective lifetimes for a surrogate chemical species of ∼2.9,
14.4, 23.1, and 28.9 hours for the 10th and 20th of August 2018. From this figure, it is clear that only for the simulations
having effective lifetimes of 23.1 hours or more, a significant fraction of the tracer emitted is present at the end of the plume as
observed in the measurements. This is further illustrated in Figure 6, depicting CHO.CHO and HCHO maps for the 10th of
August 2018. On top of these maps, contour lines are shown for the simulated air masses assuming effective lifetimes of ∼2.9,
14.4, and 28.9 hours. It is evident that in both cases the tracer distributions simulated with longer effective lifetimes better
describe the observed distribution of glyoxal and formaldehyde.





**Figure 5.** Daily maps of air masses simulated with FLEXPART for 10 and 20 of August 2018 are shown (A and B) for selected lifetimes (∼2.9, 14.4, 23.1, 28.9 hours). C) Contour plots of simulations for the same lifetimes are compared for 10 and 20 of August 2018.

Figures7-A, -B, -C present a second comparison of daily maps of glyoxal and formaldehyde VCDs with a FLEXPART tracer

having a lifetime of 28.9 hours for the period for 20th and 21st of August 2018. It is evident that again, the tracer follows the distribution of CHO.CHO and HCHO observations, similar to the first period studied (see Figure 4). However, on 20th and 21st of August, the CHO.CHO plume spreads over the ocean, where no glyoxal or formaldehyde sources are expected, until it disperses after being transported over a distance of about ∼600 km form the fires.

The observed behavior of the CHO.CHO and HCHO plumes is in contrast with the short atmospheric lifetimes resulting

from their rapid removal by photolysis and reaction with OH. In addition, CHO.CHO oligomerises and thus is a source of SOA





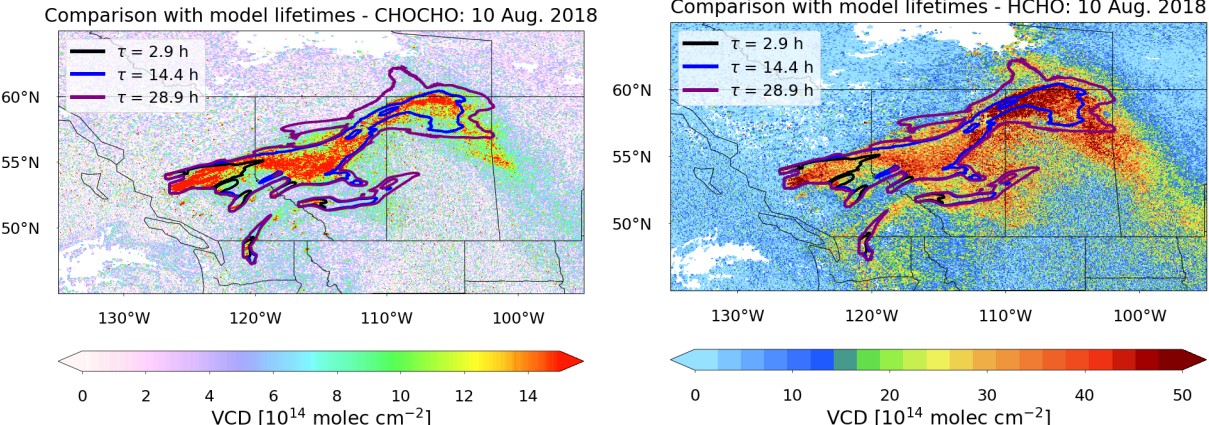

**Figure 6.** Daily maps CHO.CHO and HCHO VCD retrieved from S5P for 10th of August 2018 compared with FLEXPART tracer simulations having three different lifetimes (∼2.9, 14.4 , and 28.9 hours).

formation (Schweitzer et al., 1998; Jang et al., 2002; Liggio et al., 2005; Kroll et al., 2005; Loeffler et al., 2006; Volkamer et al., 2007; Fu et al., 2007; Myriokefalitakis et al., 2008; Stavrakou et al., 2009b, c). The simplest explanation of the observations of CHO.CHO and HCHO is that, during the fire events, both species are transported and/or produced during transport, over long distances resulting in an effective lifetime of about 28.9 hours. This implies the transport of the VOC precursors of CHO.CHO

and HCHO.

One reason for the large dispersion of the CHO.CHO and HCHO plumes is the injection of the biomass burning emissions into the free troposphere, where high wind speeds favor transport over long distances. This is a well-known effect that has also been observed for $NO_2$ in GOME-2 data (Zien et al., 2014). However, even at high wind speeds, the short lifetime of these species would result in much smaller dispersed plumes than the ones observed. There are three possible explanations for

this apparent contradiction: Reason 1: The lifetimes of CHO.CHO and HCHO could be significantly longer than expected in these biomass burning plumes. There is, however, no indication that this should be the case; on the contrary, OH levels in the biomass burning plume are expected to be enhanced (Folkins et al., 1997), leading to a reduction of the expected CHO.CHO lifetime in the unpolluted troposphere. Reason 2: There could be an efficient recycling process between the gas and aerosol phase, resulting in the observed extended effective lifetimes of CHO.CHO and HCHO. However, this reason is considered

unlikely, because there is not yet any strong evidence of HCHO being a precursor of SOA formation, and as the shape of the plumes for both trace gases is similar, a similar mechanism is expected for both. Also, evidence for the release of CHO.CHO following the formation of oligomers in the aerosol phase, is limited (Kroll et al., 2005, and reference therein).

Reason 3: The plume could contain glyoxal and formaldehyde precursors which slowly produce additional VOCs along the trajectory, resulting in an apparent increase in lifetime. In order to better assess the CHO.CHO and HCHO spatial distribution

seen on the 10th of August 2018, two additional S5P retrievals have been taken into account; the column-averaged dry air mole fractions of CO, retrieved by the algorithm described in Schneising et al. (2019), and the VCD $NO_2$ retrieved using an algorithm


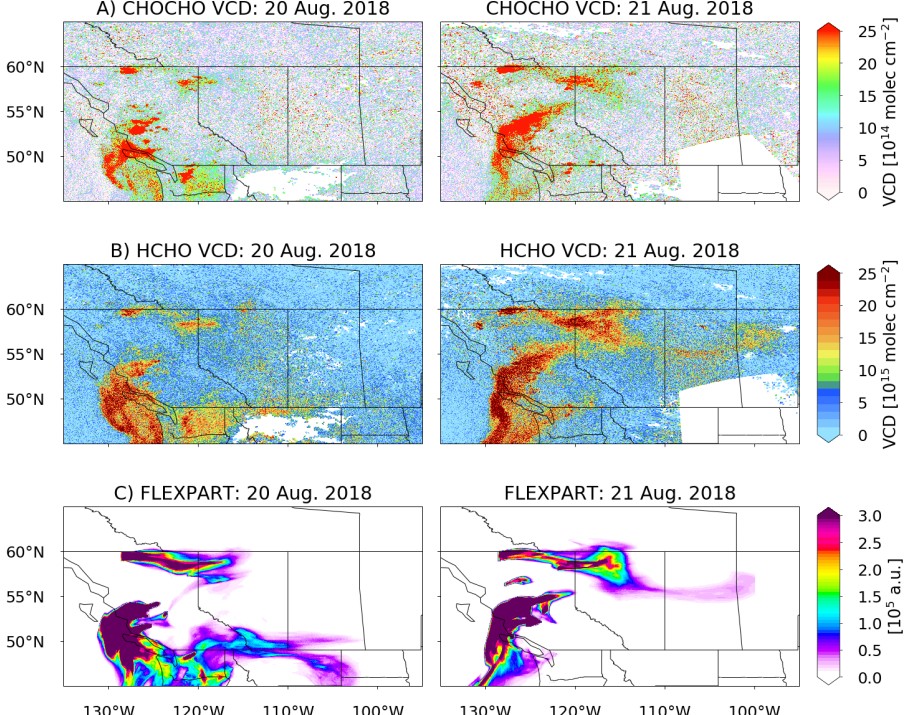

**Figure 7.** A) and B) Daily CHO.CHO and HCHO VCDs retrieved from S5P measurements for the period from 20 to 21 of August. C) Tracer distribution simulated with FLEXPART for the same period assuming a lifetime of 28.9 hours. Enhanced CHO.CHO columns spread over the ocean in a pattern similar to that simulated by the model tracer.

similar to the one described for the GOME-2 instrument (Richter et al., 2011) and using AMF calculated following the same approach as the one described before for glyoxal and formaldehyde (see section 2.3). The CO plume shows a similar spatial behavior as HCHO and CHO.CHO (see Figure 8-E). As CO is a relatively long-lived tracer of fire emissions, this is further confirmation for the fact that the VOCs and/or their precursors originate from the fires and then undergo long-range transport. $NO_2$ enhancements, in contrast, are limited to the proximity of the fire hot-spots (see Figure 8-D). This is the behavior expected for a molecule with a short atmospheric lifetime and highlights how unusual the behavior of the VOCs is. A true-color image from the Visible Infrared Imaging Radiometer Suite (VIIRS) clearly shows the distribution of aerosols (Figure 8-F), which is similar to the CHO.CHO, HCHO, and CO distributions, indicating that these species are mixed in the aerosol layer. It is interesting to note that CHO.CHO and CO follow mainly the main plume, while the HCHO distribution is more diffused and shows enhanced values also over regions where a thinner aerosol plume is visible in the VIIRS image; possibly originating from another fire as this part of the plume is not apparent in the FLEXPART simulations. As an additional criterion, the ratio of glyoxal to formaldehyde (RGF) is shown in Figure 8-C. Larger values of RGF are found close to the location of the wildfires as already reported in previous publications (Vrekoussis et al., 2010). This is an indication of enhanced primary emissions of





glyoxal from fires. Progressively lower RGF values are then found until the end of the plume suggesting either a decreasing of glyoxal production over time or a change in the fire emissions injected into the plume.

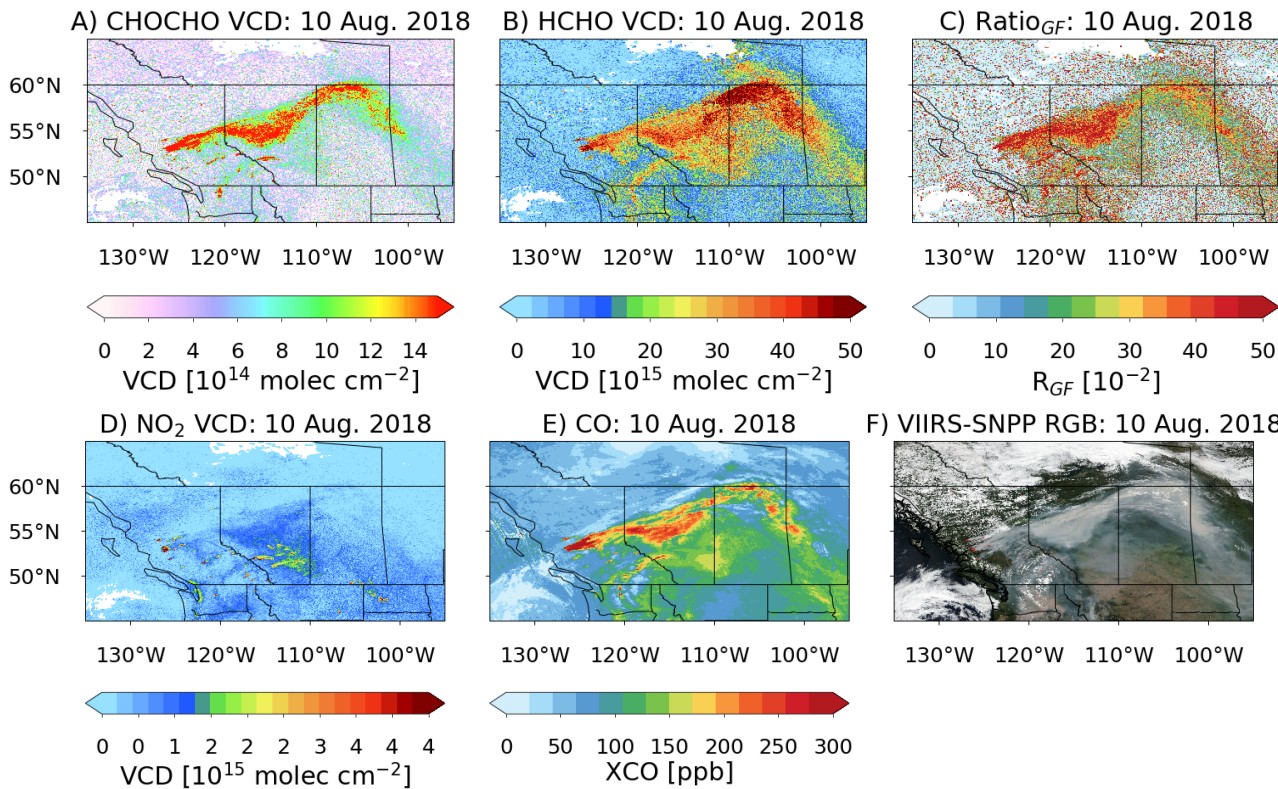

**Figure 8.** Panels A, B, D, and E show the CHO.CHO, HCHO, $NO_2$, and CO columns, respectively retrieved from S5P measurements for the 10th of August 2018. Note that CO columns are unfiltered and only represent a qualitative description of the plume. The AMFs used for CHO.CHO, HCHO, and $NO_2$ are appropriate for the biomass burning plume only. Panel C depicts the calculated CHO.CHO to HCHO ($R_{GF}$) for the same day. Panel F shows a true color image of the aerosol distribution from VIIRS, for 10th of August 2018.

The comparison of retrieved S5P columns and FLEXPART tracer simulations discussed is based on a number of simpli-fications. First of all, the observational conditions of the biomass burning plumes are complex, and aerosol scattering and absorption certainly impact on the sensitivity of the retrievals. While this is taken into account by using air mass factors for

elevated plumes positioned at altitudes derived from CALIPSO observations, there is considerable uncertainty with respect to absolute values. Aerosol loading and optical properties will vary over the plume and thus the retrieval sensitivities, and this is not modelled here. However, the differences apparent between the spatial distributions of glyoxal and $NO_2$ which are retrieved in very similar spectral regions provide evidence for the fact that measurement sensitivity is not the driver for the observed patterns. Another crucial simplification is the assumption of constant fire emissions and the proportionality of FRP and emis-

sion strength in the FLEXPART simulations. In reality, fire emissions will also depend on the type of biomass burned, the age





of the fire and the environmental conditions, and this will have an effect on the trace gas distribution along the plume which reflects both chemical transformation and the history of emissions. Modeling of this time-evolution is complex, if possible at all, and out of the scope of this study. However, the observation that both CHO.CHO and HCHO are present in the biomass burning plume after extended time periods and over long distances is robust and can only be explained by continuous release
from longer-lived precursors and/or efficient recycling processes in the plume as discussed above.

## 4   Summary and conlusions

The retrieval of formaldehyde and glyoxal total column amounts from measurements of the TROPOMI instrument onboard the Sentinel-5P satellite is reported. This enables the extension of the datasets already available from the SCIAMACHY, GOME-2, and OMI instruments, and shows the advantage of the high spatial resolution and low noise of TROPOMI for studying
specific geophysical phenomena. The satellite data show evidence for pyrogenic emissions of CHO.CHO and HCHO and their precursors during the wildfire season in summer 2018 in British Columbia, Canada. The spatial and temporal pattern of the highest retrieved CHO.CHO and HCHO columns are associated with areas having high fire radiative power, as observed in the MODIS fire data products, indicating that in these areas, pyrogenic emissions are dominant sources of CHO.CHO and HCHO. In addition to local enhancements of $NO_2$, CHO.CHO, and HCHO, an extended plume of elevated CHO.CHO
and HCHO amounts is observed on some days downwind of the fires. This finding provides evidence of either a) long-range transported VOCs, primarily emitted from the fires or b) their production from precursor species during the transport of the plume. The spatial and temporal VOC distribution observed from satellite follows the same pattern as that of CO and that simulated by the FLEXPART model, initialized by tracer emissions starting at known fire locations. Enhanced CHO.CHO and HCHO columns were found in the S5P data up to 1500 km from their sources. In order to obtain reasonable agreement between
the model results and the measurements, an effective tracer lifetime of 28.9 hours needs to be assumed in the FLEXPART simulations. This is significantly longer than the anticipated lifetimes of glyoxal and formaldehyde. The long transport of glyoxal and formaldehyde in the plume could be associated with the lifting of glyoxal and formaldehyde from the boundary layer into the free troposphere, where high wind speeds lead to rapid transport. The long apparent lifetime of CHO.CHO and HCHO could either be a real increase in atmospheric lifetime due to the specific photochemical conditions in the biomass
burning plume or as we attribute, the presence of longer-lived precursors, which are oxidized to form CHO.CHO and HCHO during transport. Based on our current knowledge, the most probable explanation would be the latter where formation of glyoxal and formaldehyde within the plume is caused by the oxidation of a mixture of longer-lived emitted VOC species (e.g. methanol, ethanol, acetylene, aromatics, glycolaldehyde, ethylene etc.), that in turn, form CHO.CHO and HCHO at different rates. Further research is needed to investigate how frequent such fire-related long-range transport events of VOCs are, what
the exact chemical mechanism of the formation within the plume is and how such events impact on ozone production and air quality downwind of the fires.



*Author contributions.* L. M. A. Alvarado, A. Richter and J. P. Burrows have prepared the manuscript with the contribution of all authors and developed the glyoxal and formaldehyde and $NO_2$ retrievals for TROPOMI measurements. M. Vrekoussis, A. Hilboll and A. B. Kalisz Hedegaard have designed and performed the FLEXPART experiment for simulation of the airmasess assuming different effective lifetimes.

O. Schneising has developed the CO retrieval and provided the CO data for the comparison with glyoxal and formaldehyde products.

*Acknowledgements.* The authors acknowledge financial support provided by the University of Bremen. Copernicus Sentinel-5P lv1 data from 2018 were used in this study. This publication contains modified COPERNICUS Sentinel data (2018). We thank the MACC team for providing the GFASv1.0 FRP and injection height products. FLEXPART simulations were conducted on the University of Bremen's HPC cluster Aether, funded by DFG. VIIRS and CALIPSO data were obtained from the NASA Langley Research Center Atmospheric Science

Data Center. This is in part preparatory work for the analysis of the data from the DFG SPP HALO EMeRGe project.



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
