# Peer review of "Unexpected long-range transport of glyoxal and formaldehyde observed from the Copernicus Sentinel-5 Precursor satellite during the 2018 Canadian wildfires"

_Atmospheric Chemistry and Physics, 2019_

## Referee Comment (RC1) · Anonymous Referee #1 · 10 Sep 2019

**Unexpected long-range transport of glyoxal and formaldehyde observed from the Copernicus Sentinel-5 Precursor satellite during the 2018 Canadian wildfires**

This paper presents satellite-derived observations of glyoxal and formaldehyde from the TROPOMI instrument, over British Columbia, Canada. Elevated column densities were associated with fire hot spots and observed over distances of up to 1500 km. Based on comparisons with FLEXPART simulations with different lifetimes, effective lifetimes of >20 hours are required to explain the observations. The authors indicate that the effective lifetimes are in contrast to the shorter expected lifetimes of these species.

My main concern with the paper is with the references to the lifetimes of glyoxal and formaldehyde. The paper does not provide adequate evidence to support the determination of atmospheric lifetimes, mainly because chemistry and deposition are not considered (and as the authors state, not within the scope of this paper). The observations of glyoxal and formaldehyde enhancements downwind of the fire hot spots are likely due to formation (and loss) processes (as the authors also note) and thus, reference to lifetimes accounting only for transport time is not appropriate.

This is a relevant paper for ACP and would be of interest to ACP readers. The paper is comprehensive, well written with clear study objectives, logically presented and articulated conclusions. The satellite-derived observations of glyoxal and formaldehyde far downwind of the fire sources are quite interesting and can stand on their own without comparison to 'expected' lifetimes.

I recommend acceptance to ACP after addressing the above comments and a few minor comments below.

L42: biomass burning includes wildfires – what is meant by indicating both?

L49: transported to 'those' regions – please clarify or reword

Intro – break into paragraphs for easier reading

L117 – any comment on the uncertainty associated with using an aerosol profile to depict the glyoxal profile?

L122 – what is meant by a homogeneous distribution? The same profile is used over the geographic region studied?

L131 – how much reduction is noise? Can this be quantified?

L132 – 'strong absorption of the latter'; of the latter not appropriate in this sentence, confusing. Remove 'of the latter' and clarify.

L150 – confusing sentence regarding lifetimes…..

L156 – 'exact emissions' ; what is meant by this? emission type (pollutant?) or emission rate?

L161 – reference should be in brackets

L218 – sentence strange; remove time of day because photon flux and OH is inherently dependent on time of day.  Also dependent on wet/dry deposition processes and other oxidants.

L220 – should be' which are discussed in detail'

Section 4 – Typo in the title 'Conclusions'

L292 – 'highest' what?  SCDs?

Table 1 caption typo, remove 'de'

---

## Referee Comment (RC2) · Anonymous Referee #2 · 25 Sep 2019

**Unexpected long-range transport of glyoxal and formaldehyde observed from the Copernicus Sentinel-5 Precursor satellite during the 2018 Canadian wildfires**

Leonardo M. A. Alvarado, Andreas Richter, Mihalis Vrekoussis, Andreas Hilboll, Anna B. Kalisz Hedegaard, Oliver Schneising, and John P. Burrows.
* * *
This paper describes TROPOMI satellite retrievals of glyoxal (CHO.CHO) and formaldehyde (HCHO) over Western Canada during the wildfire-intensive month of August 2018. Enhanced VCDs of ~14 x $10^{14}$ molec cm$^{-2}$ CHO.CHO and ~50 x $10^{15}$ molec cm$^{-2}$ HCHO are observed at wildfire locations and these enhancements appear to persist over long distances of up to 1500 km. FLEXPART tracer transport simulations using GFAS emission locations are able to reproduce the spatial distribution of enhancements if a lifetime of 20 hours or more is used.

My general suggestion for the paper is to articulate more clearly the usage of lifetimes in FLEXPART to avoid confusion. Since a full chemical transport model is not being used, (1) the model is not producing CHO.CHO and HCHO columns that can be directly compared to the observations and (2) the 'effective lifetime' does not represent the chemical/physical production/loss processes that are occurring within the large wildfire plume. Rather, the 'effective lifetime' in FLEXPART simply allows tracer particles to persist from their origin and continue to be transported. This provides a general spatial comparison to the observations. Hence the 'effective lifetime' here is a simple and useful computational proxy – but not a representation of – complex plume processes. The authors have clarified this in the Methods but it should be made more obvious to readers in other sections.

For example, the wording in the abstract suggests >24 hour lifetimes during nighttime or at high latitudes (does this refer to Canadian latitudes?) and presents 20+ hours as the FLEXPART lifetime; these are referring to the different usages described above and can be confusing.

The authors discuss the far downwind CHO.CHO and HCHO observations and suggest continued production from precursors as the likely cause, and not a physical increase in the lifetimes. It is worth noting that British Columbia is a coastal province and the presence of chlorine-initiated oxidation adds to the skepticism of >20 hr lifetimes.

This is a relevant paper for ACP. The paper is well written and the results are presented in an organized manner. The satellite retrievals of CHO.CHO and HCHO from the recently launched TROPOMI instrument are highly valuable and provide improved insight into Canadian wildfires as presented in this work. I recommend acceptance to ACP after addressing the above comments and the minor corrections below:

Line 05 – 'lifetimes'

Line 24 – order of CHO and HCHO is awkwardly changed in this sentence

Line 51 – remove comma after 'Spectroscopy'

Line 57 – remove comma after '07'

Line 73 – 'and/or'

Line 81 – capitalize 'Precursor' and remove 'of'

Line 85 – remove 'of'

Line 86 – keep formatting of dates consistent (e.g. 13 October 2017 vs. August 07 2018 in Line 57 vs. $10^{th}$ of August 2018 in Line 140, etc.)

Line 89 – 13:30 LT

Line 92 – again consider removing 'of'

Line 94/95 – keep formatting of in-text citations consistent

Line 106 – How many CHO.CHO peaks are within this range?

Line 110 – What is meant by a 'row-dependant' spectrum? Explain. Is it one background spectrum per line of latitude? The author states that a daily mean is used but if the background spectrum changes throughout the day, will this introduce significant error?

Line 110 – '…as a background spectrum (Alvarado, 2016).'

Table 1 – remove 'de' in title

Line 127 – heading should state 'HCHO retrieval from TROPOMI measurements' to match Line 93

Line 133 – 'consists'

Line 161 – full citation in brackets

Line 206 – remove comma

Line 211 – 'HCHO'

Line 220 – reword to '…which are discussed in detail in Section 3.2.'

Line 229 – reword to 'Figure 7 presents…'

Line 231 – 'However, on the $20^{th}$…'

Line 233 – 'from'

Line 252 – 'references'

Figure 8 caption – '…for the $10^{th}$ of August 2018.'

Line 286 – misspelling of 'conclusions' in heading

---

## Author Comment (AC1) · 14 Dec 2019

**Author reply to comments of anonymous Referee #1 on the manuscript**
**"Unexpected long-range transport of glyoxal and formaldehyde observed from the Copernicus Sentinel-5 Precursor satellite during the 2018 Canadian wildfires"**

Leonardo M. A. Alvarado et al.

December 13, 2019

We would like to thank the anonymous referee #1 for carefully reading our manuscript and providing valuable comments, which helped to improve the quality of our manuscript. We have answered below point by point to each comment.

We noticed a small mistake in the discussion manuscript. For the AMF calculation in the CHO.CHO and HCHO satellite retrievals, we accidentally used wrong units in the profiles used for the simulations, which created some background offset between different days. In the revised manuscript, we corrected the CHO.CHO and HCHO VCDs. This introduces only relatively small changes in the magnitude of CHO.CHO and HCHO compared to the dataset shown in the discussion manuscript and thus does not affect the interpretation of the results.
* * *
**Legend:**
- **referee comments**
- **authors comments**
* * *
This paper presents satellite-derived observations of glyoxal and formaldehyde from the TROPOMI instrument, over British Columbia, Canada. Elevated column densities were associated with fire hot-spots and observed over distances of up to 1500 km. Based on comparisons with FLEXPART simulations with different lifetimes, effective lifetimes of >20 hours are required to explain the observations. The authors indicate that the effective lifetimes are in contrast to the shorter expected lifetimes of these species.

My main concern with the paper is with the references to the lifetimes of glyoxal and formaldehyde. The paper does not provide adequate evidence to support the determination of atmospheric lifetimes, mainly because chemistry and deposition are not considered (and as the authors state, not within the scope of this paper). The observations of glyoxal and formaldehyde enhancements downwind of the fire hot spots are likely due to formation (and loss) processes (as the authors also note) and thus, reference to lifetimes accounting only for transport time is not appropriate.

We agree that full chemical simulations along the trajectory would enhance our understanding of the chemical transformation taking place as the fire emissions are transported. However, we consider the focus of this study was slightly different. We present simultaneous observations of CHO.CHO, HCHO, $NO_2$ and CO in plumes coming from wildfires. The FLEXPART simulations describe how the air masses are physically transported from the source of production, in this case, the fires. They are coupled with estimates of the lifetime of a theoretical tracer species travelling in the transported air mass. Our initial assumption, that the formaldehyde and glyoxal were produced in the fire and then transported and chemically removed, primarily by photolysis and reaction with hydroxyl radicals, clearly does not explain the observed formaldehyde and glyoxal temporal evolution. We consider that FLEXPART simulations provide an important piece of information to help us understand the behaviour of air mass plumes, as they are transported. To avoid confusion we have clarified in the text the objectives of the modelling and our use of the term "effective lifetime" in this study.

This is a relevant paper for ACP and would be of interest to ACP readers. The paper is comprehensive, well written with clear study objectives, logically presented and articulated conclusions. The satellite-derived observations of glyoxal and formaldehyde far downwind of the fire sources are quite interesting and can stand on their own without comparison to 'expected' lifetimes.
I recommend acceptance to ACP after addressing the above comments and a few minor comments below.

Thank you very much for your positive comments.

L42: biomass burning includes wildfires – what is meant by indicating both?

The sentence has been removed. What we intended to express here is that pyrogenic emissions include wildfires and agricultural fires.

L49: transported to 'those' regions – please clarify or reword
Done

Intro – break into paragraphs for easier reading
Done

L117 – any comment on the uncertainty associated with using an aerosol profile to depict the glyoxal profile?
Quantification of uncertainty associated with the assumed profile is difficult at it depends on several factors such as the geometry of observation, the presence of clouds, the altitude of aerosols, the surface albedo, etc. For this study, we consider that the most accurate approach is assuming a vertical distribution of glyoxal similar to the one measured for the aerosols. This is

because no significant contribution from other sources is expected. If there is any contribution from layers close to the ground, it is shielded by the aerosol layers and difficult to detect by satellite under the conditions of the measurements in our case study. This is because the measurement sensitivity decreases below the aerosol layer as most photons are scattered back to the satellite before they can reach these altitudes (Leitao et al., 2010). Here, a sensitivity study has been conducted assuming glyoxal profiles at different altitudes and evaluating the impact on the glyoxal AMFs. Figure 1A shows glyoxal profiles with maximum concentrations at different altitudes. Figure 1B shows the AMFs dependence with SZA for different profiles. All AMFs behave quite similar, however, for layers at higher altitude the AMFs are larger than those for a layer closer to the ground. Relative differences between AMFs were also computed using as reference the profile with maximum concentration at 2 km. The AMFs vary between 15% and 30% for small SZA but larger deviations are found for large SZA, especially for profiles with a maximum at high altitude. In general, uncertainty associated with the assumed vertical profile is one of the most significant sources of error in DOAS retrievals and can lead to uncertainties between 10 and 30% (Boersma et al., 2004; Lerot et al., 2010).

[Figure]

Figure 1: A) Glyoxal profiles peaking at different altitudes. B) Glyoxal AMFs computed using the profiles of A). C) Relative difference of AMFs for profiles at difference altitude against the AMF for the reference profile.

L122 – what is meant by a homogeneous distribution? The same profile is used over the geographic

region studied?

Yes is the short answer. We assume that the aerosols are distributed homogeneously in the whole region. For each day, the mean aerosol profile is computed as the average of all aerosol profiles measured in the region after removing cloud-contaminated pixels, and this profile is then used in the retrieval of the trace gas data.

L131 – how much reduction in noise? Can this be quantified?
The random noise in the large fitting range is about 4 times smaller than the corresponding value obtained using a smaller fit window. In the figure below, a comparison of the variation of formaldehyde slant column densities over the equatorial Pacific is shown. In this area, HCHO is mainly produced by methane oxidation and therefore assumed to be homogeneously distributed. Variations in the retrieved HCHO columns are thus taken as indication of retrieval uncertainty. The scatter obtained using a large fitting window corresponds to about $4.5 \times 10^{15}$ molec.cm$^{-2}$, while the fitting window used by Vrekoussis et al., (2010) leads to a variability of about $1.6 \times 10^{16}$ molec.cm$^{-2}$.

[Figure]

Figure 2: Distribution of S5P HCHO differential slant column densities over a clean equatorial area ocean region (5° S – 5° N, 150° – 210°) for August 2018.

L132 – 'strong absorption of the latter'; of the latter not appropriate in this sentence, confusing. Remove 'of the latter' and clarify.
The manuscript has been modified accordingly.

L150 – confusing sentence regarding lifetimes.....
The text has been modified in order to be clearer.

L156 – 'exact emissions'; what is meant by this? emission type (pollutant?) or emission rate?
Here, we meant "emission rate", which has been clarified in the revised manuscript.

L161 – reference should be in brackets
Done

**References**
Leitão, J., Richter, A., Vrekoussis, M., Kokhanovsky, A., Zhang, Q. J., Beekmann, M., and Burrows, J. P.: On the improvement of NO2 satellite retrievals – aerosol impact on the airmass factors, Atmos. Meas. Tech., 3, 475–493, doi:10.5194/amt-3-475-2010, 2010.

Boersma, K. F., Eskes, H. J., and Brinksma, E. J.: Error analysis for tropospheric NO2 retrieval from space, J. Geophys. Res., 109, D04311, doi:10.1029/2003JD003962, 2004.

Lerot, C., Stavrakou, T., De Smedt, I., Müller, J.-F., and Van Roozendael, M.: Glyoxal vertical columns from GOME-2 backscattered light measurements and comparisons with a global model, Atmos. Chem. Phys., 10, 12 059–12 072, doi:10.5194/acp-10-12059-2010, 2010.

Vrekoussis, M., Wittrock, F., Richter, A., and Burrows, J. P.: GOME-2 observations of oxygenated VOCs: what can we learn from the ratio glyoxal to formaldehyde on a global scale?, Atmos. Chem. Phys., 10, 10 145–10 160,  doi:10.5194/acp-10-10145-2010, 2010.

---

## Author Comment (AC2) · 14 Dec 2019

**Author reply to comments of anonymous Referee #2 on the manuscript**
**"Unexpected long-range transport of glyoxal and formaldehyde observed from the Copernicus Sentinel-5 Precursor satellite during the 2018 Canadian wildfires"**

Leonardo M. A. Alvarado et al.

December 13, 2019

We would like to thank the anonymous referee #2 for carefully reading our manuscript and for providing valuable comments, which helped to improve the quality of our manuscript. We have answered below point by point to each comment.

We noticed a small mistake in the discussion manuscript. For the AMF calculation in the CHO.CHO and HCHO satellite retrievals, we accidentally used wrong units in the profiles used for the simulations, which created some background offset between different days. In the revised manuscript, we corrected the CHO.CHO and HCHO VCDs. This introduces only relatively small changes in the magnitude of CHO.CHO and HCHO compared to the dataset shown in the discussion manuscript and thus does not affect the interpretation of the results.

**Legend:**
- **referee comments**
- **authors comments**

This paper describes TROPOMI satellite retrievals of glyoxal (CHO.CHO) and formaldehyde (HCHO) over Western Canada during the wildfire-intensive month of August 2018. Enhanced VCDs of ~$14 \times 10^{14}$ molec cm$^{-2}$ CHO.CHO and ~$50 \times 10^{15}$ molec cm$^{-2}$ HCHO are observed at wildfire locations and these enhancements appear to persist over long distances of up to 1500 km. FLEXPART tracer transport simulations using GFAS emission locations are able to reproduce the spatial distribution of enhancements if a lifetime of 20 hours or more is used.

My general suggestion for the paper is to articulate more clearly the usage of lifetimes in FLEXPART to avoid confusion. Since a full chemical transport model is not being used, (1) the model is not producing CHO.CHO and HCHO columns that can be directly compared to the observations and (2) the 'effective lifetime' does not represent the chemical/physical production/loss processes that are occurring within the large wildfire plume. Rather, the 'effective lifetime' in FLEXPART simply allows tracer particles to persist from their origin and continue to be transported. This provides a general spatial comparison to the observations. Hence the 'effective lifetime' here is a simple and useful computational proxy – but not a representation of – complex plume processes. The authors have clarified this in the Methods but it should be made more obvious to readers in other sections.

For example, the wording in the abstract suggests >24 hour lifetimes during night-time or at high latitudes (does this refer to Canadian latitudes?) and presents 20+ hours as the FLEXPART lifetime; these are referring to the different usages described above and can be confusing.

The authors discuss the far downwind CHO.CHO and HCHO observations and suggest continued production from precursors as the likely cause, and not a physical increase in the lifetimes. It is worth noting that British Columbia is a coastal province and the presence of chlorine-initiated oxidation adds to the skepticism of >20 hr lifetimes.

We thank the reviewer for the comment. We agree that this study does not try to determine the current chemical lifetimes of these species, and the simulations are only used to describe how they are physically transported from the source of production. The estimated lifetime over which these species are observed in the atmosphere corresponds to the assumption of a simple exponential first-order decay which is not representative for the complex chemistry in the plume. We have clarified in the text what we mean by this effective lifetime in our study to avoid confusion. However, we believe that FLEXPART simulations provide an important piece of information for understanding the behaviour of the plume until it is dispersed.
Regarding the latitudes in the manuscript, these refer to Canadian latitudes. The lifetimes of VOC depend on the season and these are connected to OH variability, and thus to photolysis as well. Globally, OH significantly decreases for latitudes larger than 45°N (Lelieveld et al., 2016), which corresponds to Canadian latitudes, and thus we expect longer lifetimes for VOCs over these latitudes.

This is a relevant paper for ACP. The paper is well written and the results are presented in an organized manner. The satellite retrievals of CHO.CHO and HCHO from the recently launched TROPOMI instrument are highly valuable and provide improved insight into Canadian wildfires as presented in this work. I recommend acceptance to ACP after addressing the above comments and the minor corrections below:
Thank you very much for your positive comments.

Line 05 – 'lifetimes'
Done

Line 24 – order of CHO and HCHO is awkwardly changed in this sentence
It has been rearranged in the revised manuscript.

Line 51 – remove comma after 'Spectroscopy'
Done

Line 57 – remove comma after '07'
Done

Line 73 – 'and/or'
Done

Line 81 – capitalize 'Precursor' and remove 'of'
Done

Line 85 – remove 'of'
Done

Line 86 – keep formatting of dates consistent (e.g. 13 October 2017 vs. August 07 2018 in Line 57 vs. 10th of August 2018 in Line 140, etc.)
The text has been modified in order to be consistent.

Line 89 – 13:30 LT
The manuscript has been modified accordingly.

Line 92 – again consider removing 'of'
Done

Line 94/95 – keep formatting of in-text citations consistent
The manuscript has been modified accordingly.

Line 106 – How many CHO.CHO peaks are within this range?
In the fitting range used in this study, five glyoxal absorption peaks can be found (see Figure 2A in the manuscript), including the strongest absorption band of glyoxal.

Line 110 – What is meant by a 'row-dependant' spectrum? Explain. Is it one background spectrum per line of latitude? The author states that a daily mean is used but if the background spectrum changes throughout the day, will this introduce significant error?
Here, a daily background spectrum is computed by averaging over the whole latitude range (50° S – 50° N) for each across-track viewing direction individually. Thus, one background spectrum is used per viewing direction (450 spectra). This approach is taken to minimise small across-track dependent differences of the TROPOMI measurements.
Ideally, a solar irradiance measurement would be used as a background spectrum in the analysis. However, for weak absorbers such as glyoxal and formaldehyde, use of a daily Earth shine background derived by averaging measurements over the Pacific has proven to reduce noise and offsets in the data. Day to day changes of this background spectrum are small and are therefore expected to introduce only small uncertainties. However, over longer time periods (weeks and months), instrumental drift may induce changes in trace gas columns if the background spectrum is not based on recent measurements. Use of a daily background spectrum resulted in a significant reduction of instrumental noise similar to that demonstrated in previous studies (Schönhardt et al. 2008, De Smedt et al., 2008, Anand et al., 2015, Alvarado, 2016).

Line 110 – '...as a background spectrum (Alvarado, 2016).'
Done

Table 1 – remove 'de' in title
Done

Line 127 – heading should state 'HCHO retrieval from TROPOMI measurements' to match Line 93
Done

Line 133 – 'consists'
Done

Line 161 – full citation in brackets
Done

Line 206 – remove comma
Done

Line 211 – 'HCHO'
Done

Line 220 – reword to '...which are discussed in detail in Section 3.2.'
The manuscript has been reworded accordingly.

Line 229 – reword to 'Figure 7 presents...'
The manuscript has been reworded accordingly.

Line 231 – 'However, on the 20th...'
Done

Line 233 – 'from'
Done

Line 252 – 'references'
Done

Figure 8 caption – '...for the 10th of August 2018.'
Done

Line 286 – misspelling of 'conclusions' in heading
This has been corrected in the manuscript.

**References**
Anand, J. S., Monks, P. S., and Leigh, R. J.: An improved retrieval of tropospheric NO2 from space over polluted regions using an Earth radiance reference, Atmos. Meas. Tech., 8, 1519–1535, 2015.

Alvarado, L. M. A., Richter, A., Vrekoussis, M., Wittrock, F., Hilboll, A., Schreier, S. F., and Burrows, J. P.: An improved glyoxal retrieval from OMI measurements, Atmos. Meas. Tech., 7, 4133–4150, 2014.

De Smedt, I., Müller, J.-F., Stavrakou, T., van der A, R., Eskes, H., and Van Roozendael, M.: Twelve years of global observations of formaldehyde in the troposphere using GOME and SCIAMACHY sensors, Atmos. Chem. Phys., 8, 4947–4963, 2008.

Lelieveld, J., Gromov, S., Pozzer, A., and Taraborrelli, D.: Global tropospheric hydroxyl distribution, budget and reactivity, Atmos. Chem. Phys., 16, 12477–12493, acp-16-12477-2016, 2016.

Schönhardt, A., Richter, A., Wittrock, F., Kirk, H., Oetjen, H., Roscoe, H. K., and Burrows, J. P.: Observations of iodine monoxide columns from satellite, Atmos. Chem. Phys., 8, 637–653, 2008.